# Do Urban Hedgehogs (*Erinaceus europaeus*) Represent a Relevant Source of Zoonotic Diseases?

**DOI:** 10.3390/pathogens12020268

**Published:** 2023-02-07

**Authors:** Catarina Jota Baptista, Paula A. Oliveira, José M. Gonzalo-Orden, Fernanda Seixas

**Affiliations:** 1Department of Veterinary Sciences, School of Agrarian and Veterinary Sciences (ECAV), University of Trás-os-Montes and Alto Douro (UTAD), Quinta de Prados, 5001-801 Vila Real, Portugal; 2Centre for Research and Technology of Agro-Environmental and Biological Sciences (CITAB), Inov4Agro, University of Trás-os-Montes and Alto Douro (UTAD), Quinta de Prados, 5000-801 Vila Real, Portugal; 3Institute of Biomedicine (IBIOMED), University of León, 24071 León, Spain; 4Veterinary and Animal Research Center (CECAV), AL4Animals, University of Trás-os-Montes and Alto Douro (UTAD), Quinta de Prados, 5000-801 Vila Real, Portugal

**Keywords:** hedgehog, wildlife, zoonotic pathogens, habitat, one health

## Abstract

Urban fauna is defined as animal species that can live in urban environments. Several species, including the western-European hedgehog (*Erinaceus europaeus*), have now been identified as part of this urban fauna, becoming permanent residents of parks and gardens in different cities across Europe. Due to the importance that this phenomenon represents for zoonotic disease surveillance, several authors have been conducting zoonotic agents’ surveys on hedgehog. The aim of this study is to compare zoonotic diseases’ prevalence in hedgehogs in urban environments with those from more rural areas. A systematic review with meta-analysis of twelve studied of zoonotic diseases’ (in urban and rural areas of Europe) was therefore conducted for this purpose. Fifteen different zoonoses have been assessed in urban environments and six in rural areas. *Anaplasma phagocytophilum* was the most prevalent zoonotic agent found in urban habitats (96%). Dermatophytosis shows statistically significant differences between locations (*p*-value < 0.001), with a higher prevalence in urban Poland (55%; n = 182). Our results suggest further research and a standardized monitoring of different hedgehog populations are essential to understanding the epidemiology of several zoonotic pathogens in different habitat types (urban, rural, natural, industrial, etc.) and preventing possible disease outbreaks.

## 1. Introduction

During the last decades, the global human population has grown substantially worldwide, fragmenting, occupying, or polluting natural habitats, which seriously affects the survival of many species [1]. Densely populated cities have now infiltrated almost every country in the world, spreading to the outskirts and invading natural areas and habitats [2]. Consequently, wild species have had to adapt to new environments (such as farms, family gardens, or urban parks), which may involve new shelters, food availability, predators, competitors, and even pathogens to deal with. On the other hand, their close contact with humans represents a novel source of important infectious diseases for humans that may lead to new outbreaks and epidemics. For these reasons, this phenomenon deserves attention and monitoring from the scientific community [3,4].

Many studies have shown that species-habitat relations and ecological niches are already well-established in urban areas, just like they are in purely natural environments, leading to the permanent presence of several wild animal populations [5]. The human perception of this trend varies depending on the species and the geographical area. Some cases are first reported as positive opportunities for close contact with nature. Some positive examples include when wild turkeys roamed the paths of Harvard University [6] or when people began to build artificial refuges for hedgehogs in private gardens [7]. However, other situations are clear examples of human-wildlife conflicts, such as wild boars invading private properties near Barcelona (Spain), sightings of bears near villages in Romania, and several cases of physical aggression and bites from raccoons [6,8].

From a public health perspective, close contact opportunities between new species (in this case, humans and urban wildlife) are always opportunities for cross-species transmissions and new outbreaks of zoonotic diseases. Therefore, they should be approached as a One Health concern [9,10]. One Health is a transdisciplinary concept focused on dealing with the health of humans, animals, and the environment simultaneously, recognizing their intimate relationship. Therefore, the need to include wildlife research in the One Health approach for it to be truly effective is easily understandable [10].

The western-European hedgehog (*Erinaceus europaeus*) is a nocturnal, insectivorous small mammal classified as “Least Concern” by the International Union for Conservation of Nature (IUCN), broadly distributed in continental Europe and also present in the Azores Islands and New Zealand [11]. Besides rural and natural areas, hedgehogs have been found more frequently in green areas and city parks in high-density urban centers, demonstrating adaptability and resilience to different resources [12]. However, in some countries (such as the UK), urban predators such as domestic dogs, foxes (*Vulpes vulpes*), and badgers (*Meles meles*) have been negatively affecting the hedgehog population [13]. Nevertheless, hedgehogs can be assumed to be stable in most European countries [14,15]. The diet of a hedgehog is usually composed of earthworms, slugs, and insects; however, they may also eat small prey (as mice, snakes, and birds), eggs, and pet food offered by people [16], which enhances close contact with people.

The population of hedgehogs may reveal a significant prevalence of zoonotic vector-borne diseases (such as *Anaplasma phagocytophilum* or *Borrelia burgdorferi* sensu lato (s.l.)) that may infect other hosts through ticks and fleas [16,17,18,19,20]. They may also carry skin dermatophytes (such as *Trichophyton erinacei*) [21,22], and a variety of bacteria and protozoa (*Salmonella* spp., *Leptospira* spp., *Giardia* spp., *Cryptosporidium* spp., etc.) [23,24,25,26,27], which may be transmitted through direct contact or contact with excrements to humans and animals and accumulate in the environment. Furthermore, hedgehogs are hosts of a group of coronaviruses (*Erinaceus* coronaviruses) [28,29,30], whose zoonotic potential is still unclear.

The aim of this review is (1) to compare different zoonoses assessments in *E. europaeus* from rural and urban areas; (2) to discuss if urban hedgehogs are more often reservoirs of zoonotic diseases (and why); and, finally, if they should be priorities for monitoring urban outbreaks of these diseases.

## 2. Materials and Methods

### 2.1. Literature Search and Inclusion Criteria

To obtain the necessary data for this study, we used the search tools Scopus, PubMed, and Google Scholar. At first, the keywords hedgehog; Europe (or European); and zoonotic (or zoonosis) were used to obtain a primary group of articles. All study types were included (articles, reports, book chapters, etc.). All of the different diagnostic methods and laboratory tests used by the authors were considered equally valid for this study’s purpose. All pathogens mentioned in the literature as “potentially zoonotic” (whose transmission to humans has not been completely determined or understood) were also included in our database. This systematic review was designed and conducted according to the PRISMA guidelines. The flow diagram is available in the Appendix A.

### 2.2. Exclusion Criteria

Then, exclusion criteria were applied and comprised the following principles: studies focusing on other hedgehog species (rather than *E. europaeus*); studies that did not specify the study location (or its classification as urban, rural, or other); studies that did not indicate the prevalence of the disease in hedgehogs; and studies unrelated to the subject.

Therefore, 12 papers were included, interpreted, and compared in our study. Some report the prevalence of more than one disease or geographical location, resulting in a total of 28 zoonotic disease assessments.

### 2.3. Data Extraction

Data were collected regarding the zoonotic disease, disease type (fungal, parasitic, viral, or bacterial), study location, location type (urban or rural), country, number of animals tested (n), the prevalence of disease (*p*), the author, and the year of publication.

### 2.4. Publication Bias

In the present study, publication bias did not apply as most analyses were performed within the same study or between two studies [31].

### 2.5. Comparing Prevalences—Statistical Analysis

Prevalence rates of *B. burgdorferi* (s.l.) were compared between six locations in Central Europe [19]. Considering dermatophytosis, prevalences obtained from different studies and locations were also compared, taking study-level effects into account. Following previously described methods by Kutzke [32], the meta-analysis and statistical analyses were designed to compare the prevalence between geographical areas or studies. Microsoft Office 365 Excel and IMB SPSS Statistics 27 were used to produce graphs and perform statistical tests (the Chi-Square test). A critical *p*-value of 0.05 and a confidence interval of 95% (CI 95%) were considered statistically significant.

## 3. Results

Table 1 and Table 2 present the zoonotic agents’ prevalences found in hedgehogs (a total of seventeen) from urban and rural areas, respectively. Fifteen different zoonoses were assessed for urban environments and only six for rural areas.

### 3.1. Bacterial Diseases

A total of seven bacterial zoonoses with different prevalences have been reported in hedgehogs, in both rural and urban environments.

*B. burgdorferi* sensu lato (s.l.) was assessed in both habitats, urban and rural (Table 1 and Table 2). Considering the different rural habitats, prevalence varied between 15% and 33%, while in urban areas these values varied from 10% and 13% [19]. The prevalence and its 95% CI for each location are illustrated in Figure 1. According to the Chi-square test, there are no significant differences between the prevalence found in the six different geographical areas (*p*-value = 0.61) (Appendix B; Table A1).

*Salmonella* spp. was only assessed in urban hedgehogs, revealing a prevalence (*p*) of 10% (n = 90) in the Netherlands [34], and 57% (n = 37) in Finland [33]. On the other hand, *E. coli* (ESC-resistant) (*p* = 71%, n = 90) and *Campylobacter* spp. (*p* = 1%, n = 90) were also detected in the same urban sites in the Netherlands. *Pasteurella multocida*, *Corynebacterium ulcerans*, and *Staphylococcus intermedius* were found in the city of Joensuu, Finland (*p* = 8%, n = 37; *p* = 14%, n = 37; and *p* = 5%, n = 37, respectively).

Outside continental Europe, hedgehogs were found as hosts of *Leptospira interrogans* s.l. (*p* = 27%, n = 11) in the Azores, Portugal.

### 3.2. Fungal Diseases

The prevalence of dermatophytes (mostly from the genus *Trichophyton*) was assessed in rural (in France and the UK) and urban habitats (in Poland and the UK) (Table 1 and Table 2). Urban hedgehogs seemed to show higher prevalence of these pathogens, 17% (n = 77) and 55% (n = 182), compared to rural hedgehogs, 8% (n = 24) and 9% (n = 57). A comparison of prevalence and 95% CI is illustrated in Figure 2. According to the Chi-square test, there is a significant difference between the prevalence found in those four different geographical areas (*p*-value < 0.001) (Appendix B; Table A2).

### 3.3. Parasitic Diseases

More parasitic diseases have been reported in urban areas. Vectors (such as fleas and ticks) and vector-borne parasites (*Anaplasma phagocytophilum*) had the highest prevalences, considering all the recorded pathogens. In Ulm (Germany), ticks (as *Ixodes* spp.) and fleas (as *Archeopsylla erinacei*) were present in 88% and 89% of the hedgehogs (n = 56), respectively [39]. *A. phagocytophilum* was assessed in 112 hedgehogs from three different cities in the Czech Republic with a very high prevalence (96%) [37]. Considering gastrointestinal parasites, such as *Giardia* spp., they were more prevalent in rural areas (33%) [26], compared to urban zones (11%) [34].

### 3.4. Viral Diseases

Regarding potentially zoonotic viruses, the prevalence of *Erinaceus* coronaviruses (EriCoVs) was higher in rural areas of Italy (*p* = 80%; n = 5) than in urban regions of the same country (*p* = 53%; n = 19).

## 4. Discussion

In this study, seventeen zoonotic agents and their prevalence were assessed in different locations, and surveys of *E. europaeus* were performed. While fifteen pathogens have been assessed in urban environments, only six were identified in rural areas. Isolated from other data, this finding may give the first impression that hedgehogs from urban environments are infected with a wider range of zoonotic pathogens. In addition, *A. phagocytophilum* was the most prevalent (*p* = 96%) and was found in hedgehogs from urban centres. In a general sense, pathogen transmission can sometimes increase among urban-adapted wild hosts compared to those living outside the city limits [3]. However, differences in the number of animals, geographical areas, methods, or infectious agent epidemiology may influence this. Furthermore, urban hedgehogs may be more frequently found (and, consequently, be part of an assessment) due to their proximity to people.

*B. burgdorferi* (s.l.), the agent responsible for Lyme disease, is mentioned as a good example of how the “dilution effect” concept is applied to urban diseases transmitted by wildlife. The dilution effect happens when there are abundant and diverse host species. If vectors (such as ticks) feed on multiple host species (in different proportions), contracting, amplifying, and transmitting the pathogen, the transmission is lower. The reverse situation could occur in urbanised areas, with an expectedly low host diversity leading to increased transmission. A study in a suburban northeastern area of the USA with low mammalian biodiversity showed that the high abundance of the white-footed mouse (*Peromyscus leucopus*), a very important reservoir of *B. burgdorferi* (s.l.), was linked to more infected vectors and mammalian hosts, including humans [3,42,43,44,45]. Therefore, it would be expected to find differences between the urban and rural locations. However, our results did not show statistical differences in disease prevalence between the six locations in Central Europe studied by Skuballa et al. [19]. Lake Constantine presents a very wide 95% CI due to the reduced number of hedgehogs assessed (n = 6). For *B. burgdorferi* (s.l.) and other vector-borne zoonoses, it is also important to consider the prevalence of each vector species (such as *Ixodes* spp.), regardless of study location or location type. Some authors suggested that *I. ricinus* is more important in maintaining *B. burgdorferi* (s.l.) infection in hedgehog populations than *I. hexagonus*, which is responsible for lower infection levels. In a study in the UK, no infected hedgehogs were found due to the absence of *I. ricinus* in suburban gardens [19].

Regarding other vector-borne zoonoses, such as *A. phagocytophilum*, a high prevalence was found in urban centers of the Czech Republic (*p* = 96%; n = 112), even though we do not have a rural assessment of *E. europaeus* to compare this value with. Notwithstanding, close to these locations, in South-Western Slovakia, urban and natural prevalences of *A. phagocytophilium* in ticks and rodents were compared. Our results revealed that urban *I. ricinus* populations were infected with *A. phagocytophilum* at a higher rate, but the results were inconclusive regarding rodents, suggesting they are not the main reservoirs of this pathogen [46]. In Brazil, domestic dogs from urban areas have been associated with higher infection rates of ticks and tick-borne agents (such as *A. phagocytophilum*) compared to rural dogs [47].

The nomenclature of *Salmonella* spp. is often complex due to the variety of species, subspecies, and phage types. The epidemiology and other aspects of salmonellosis are equally complicated due to the variety of hosts and clinical presentations [23]. Nevertheless, wildlife seems to also play a role in its presence or presentation to humans. Regarding *E. europaeus*, *Salmonella* spp. was only assessed in urban environments, revealing a prevalence of 10% (n = 90) in the Netherlands [34] and 57% (n = 37) in Finland [33]. In contrast, both urban and rural capybaras (*Hydrochoerus hydrochaeris*) revealed positive results (in bacterial culture and PCR) for *Salmonella* spp., suggesting their presence in both urban and rural fauna [48].

Figure 2 and Table A2 suggest a higher prevalence of dermatophytosis in urban hedgehogs. Even though the prevalence is especially distinctive in urban hedgehogs from Poland [35], in the UK, urban hedgehogs also present a high prevalence of the disease compared to rural individuals from the same country [36]. Dermatophytes (*Trichophyton* spp., *Microsporum* spp., *Epidermophytonare* spp., etc.) are ubiquitarian fungi that may be found in urban and rural environments and are responsible for a dermatologic disease called ringworm disease [49,50]. Dermatophytes can be found and transmitted through soils or via direct or indirect contact with infected humans or animals that may show clinical signs or be asymptomatic carriers [35,51]. According to a study, urban soils and surfaces seemed to be more suitable reservoirs for almost all dermatophytes [50]. On the other hand, another study suggested that human ringworm in rural areas is mainly of animal origin, while in urban areas, it is mainly contracted from other humans or pets (dogs and cats) [49]. In rescued hedgehogs, *T. erinacei* was the most frequently registered (94.6%), followed by *T. mentagrophytes*. *T. mentagrophytes* has been reported from a wide spectrum of hosts, including domestic species [52]. Le Barzic et al. [52] highlight the importance of asymptomatic hedgehogs in this disease’s transmission to humans, since one caretaker and one clinician who participated in their study developed ringworm-associated lesions, likely associated with the manipulation of hedgehogs.

During the last twenty years, betacoronaviruses (BetaCoVs) have been demonstrating their zoonotic potential. The Middle East respiratory syndrome coronavirus (MERS-CoV) was one of the first globally recognized examples. More recently, there was another obvious example, the severe acute respiratory syndrome coronavirus (SARS-CoV-2). MERS-related CoVs in hedgehogs have been reported in France [29], Germany [28], and the UK [53], suggesting that *E. europaeus* represents a wild reservoir of betacoronaviruses, namely EriCoVs [30,40,53]. Despite their considerable prevalence, viral diseases are less studied than other disease types when it comes to comparing urban and rural habitats. Delogu et al. [40] compared several urban and rural regions of Italy and found no statistical difference between the rural and urban prevalences of EriCoVs. These authors suggested that it might happen due to the circulation of viruses in rescue centers and to the consistent intersections of home ranges. These authors also reinforced the potential of *E. europaeus* as a maintenance host, bridge host, or dead-end host in the epidemiology of several emerging agents (including viruses), due to (1) their feeding and ecological habits, (2) high population densities, and (3) their common intra- and interspecies interactions (with other species, including humans) [40].

Certainly, multiple demographic, ecological and geographical aspects from each location contribute to the differences between rural and urban areas (e.g., population density, number of green areas and refugia, presence of other species, etc.). Further epidemiological studies and urban wildlife monitoring assessments may contribute to characterizing them in detail.

With a few exceptions, including *B. burgdorferi* (s.l.) and dermatophytosis, the prevalence of most diseases presented in this study was not assessed in both environmental types or more than one or two locations, which makes it impossible or inaccurate to compare them.

## 5. Conclusions

Our findings on the surface level suggest that urban hedgehogs may seem to carry more zoonotic agents (and be more important reservoirs) than those from rural areas. The differences in the variety of agents detected in both environments and the higher prevalence of dermatophytes in urban areas may support this idea. Nevertheless, this is not valid or obvious for all pathogens, and currently there is insufficient data in the literature to support this hypothesis. Further research, assessing several agents, using the same methodology, and comparing animals from rural and urban areas in the same countries/regions is necessary to determine if urban hedgehogs deserve special attention regarding zoonoses outbreaks.

Urban centers (especially when densely populated or with poor sanitary conditions) represent ideal places for infectious disease dissemination. Therefore, zoonotic disease assessments (with standardized methods) of urban fauna are crucial to recognize and prevent potential outbreaks of One Health importance.

## Figures and Tables

**Figure 1 pathogens-12-00268-f001:**
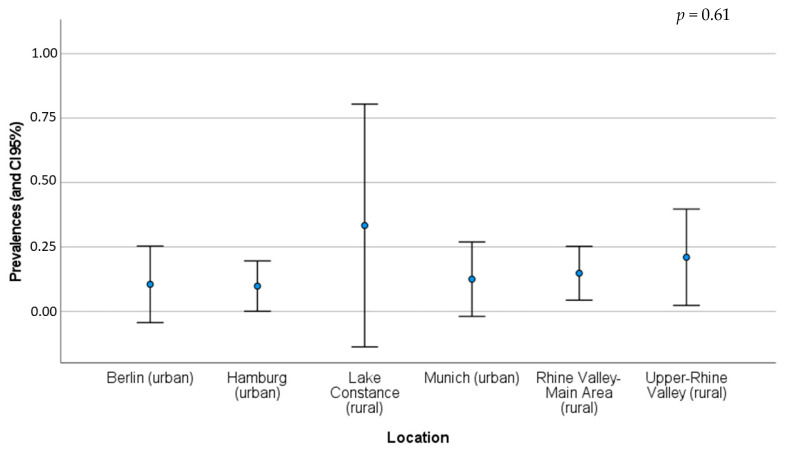
Detected prevalence (blue point) and CI 95% of *B. burgdorferi* (s.l.) for each location and location type in Central Europe (see Table 1 and Table 2 for more details about each location) [19].

**Figure 2 pathogens-12-00268-f002:**
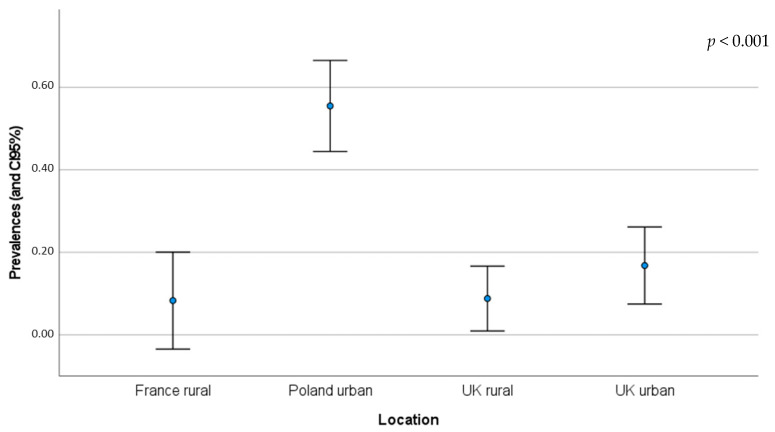
Detected prevalence (blue point) and 95% CI of dermatophytosis for each location and location type (check Table 1 and Table 2 for more details about each location) (*p*-value < 0.001) [35,36].

**Table 1 pathogens-12-00268-t001:** Hedgehogs’ zoonotic disease prevalence, categorized based on sampled urban locations.

	Agent	*p* (%) ^1^	n	City/Local	Country	Ref
Bacteria	*Borrelia burgdorferi* s.l.	10%	41	Hamburg	Germany	[19]
	*Borrelia burgdorferi* s.l.	11%	19	Berlin	Germany	[19]
	*Borrelia burgdorferi* s.l.	13%	24	Munich	Germany	[19]
	*Salmonella* spp.	57%	37	Joensuu	Finland	[33]
	*Salmonella* spp.	10%	90	Flevoland, Gelderland, Noord-Holland, Utrecht, and Zuid-Holland	The Netherlands	[34]
	*Pasteurella multocida*	8%	37	Joensuu	Finland	[33]
	*Corynebacterium ulcerans*	14%	37	Joensuu	Finland	[33]
	*Staphylococcus intermedius*	5%	37	Joensuu	Finland	[33]
	*Campylobacter* spp.	1%	90	Flevoland, Gelderland, Noord-Holland, Utrecht, and Zuid-Holland	The Netherlands	[34]
	*E. coli* (ESC-resistant)	71%	90	Flevoland, Gelderland, Noord-Holland, Utrecht, and Zuid-Holland	The Netherlands	[34]
Fungi	Dermatophytes	55%	182	(not specified—six cities with more than 100,000 inhabitants)	Poland	[35]
	Dermatophytes	17%	77	Berkshire, Cardiganshire, Devon, Essex, Hertfordshire, Leicestershire, London, Norfolk, Somerset, Surrey, and Yorkshire	UK	[36]
Parasites	*Anaplasma phagocytophilum*	96%	112	Brno, Prague, and České Budějovice	Czech Republic	[37]
	*Toxoplasma gondii*	19%	26	Brno, Prague, and České Budějovice	Czech Republic	[38]
	Fleas	89%	56	Ulm	Germany	[39]
	Ticks	88%	56	Ulm	Germany	[39]
	*Giardia* spp.	11%	90	Flevoland, Gelderland, Noord-Holland, Utrecht, and Zuid-Holland	The Netherlands	[34]
	*Cryptosporidium* spp.	9%	90	Flevoland, Gelderland, Noord-Holland, Utrecht, and Zuid-Holland	The Netherlands	[34]
Viruses	*Erinaceus* coronaviruses	53%	19	Bologna, Casalecchio, Minerbio, Lugo, Copparo, Imola, and Granarolo	Italy	[40]

^1^ Detected prevalence (%).

**Table 2 pathogens-12-00268-t002:** Hedgehogs’ zoonotic disease prevalence, categorized based on sampled rural locations.

	Agent	*p* (%) ^1^	n	City/Local	Country	Ref
Bacteria	*Borrelia burgdorferi* s.l.	33%	6	Lake Constance	Germany, Austria, Switzerland.	[19]
	*Borrelia burgdorferi* s.l.	21%	24	Upper-Rhine Valley	Germany, Switzerland.	[19]
	*Borrelia burgdorferi* s.l.	15%	54	Rhine Valley-Main Area	Germany	[19]
	*Leptospira interrogans* s.l.	27%	11	Azores (São Miguel)	Portugal	[27]
Fungi	Dermatophytes	9%	57	Near Berkshire, Cardiganshire, Devon, Essex, Hertfordshire, Leicestershire, London, Norfolk, Somerset, Surrey, and Yorkshire	UK	[36]
	Dermatophytes	8%	24	Pas de Calais	France	[36]
Parasites	*Giardia* spp.	33%	6	Southeastern North Island	New Zealand	[26]
Viruses	*Erinaceus* coronaviruses	80%	5	Bentivoglio, Budrio, and Conna	Italy	[40]
	*Belerina* virus	39%	147	Merelbek, Maldere, and Opglabbe	Belgium	[41]

^1^ Detected prevalence (%).

## Data Availability

Not applicable.

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
