# Peer review of "Do Urban Hedgehogs (Erinaceus europaeus) Represent a Relevant Source of Zoonotic Diseases?"

_pathogens, 2023, doi:10.3390/pathogens12020268_

Round 1

Reviewer 1 Report

The manuscript " Do urban hedgehogs (Erinaceus europaeus) represent a relevant source of zoonotic diseases?" compares zoonotic diseases prevalence in hedgehogs from urban environments with those from more rural areas. However some points need to be addressed. 

The authors showed the prevalence of dermatophytosis in urban areas compared to rural environments, however the cities characteristics with highest prevalance were not addressed, such as population density and presence of green areas and city parks. In addition, studies of E. europaeus prevalence in domestic species in these areas were also unrelated.

Author Response

The manuscript " Do urban hedgehogs (Erinaceus europaeus) represent a relevant source of zoonotic diseases?" compares zoonotic diseases prevalence in hedgehogs from urban environments with those from more rural areas. However some points need to be addressed. 

We would like to thank this reviewer for taking time reviewing our manucript and for the positive and constructive feedback regarding it.

The authors showed the prevalence of dermatophytosis in urban areas compared to rural environments, however the cities characteristics with highest prevalance were not addressed, such as population density and presence of green areas and city parks. In addition, studies of E. europaeus prevalence in domestic species in these areas were also unrelated.

We understand the reviewer suggestion on mentioning the characteristics of the urban areas (green parks, population density...) regarding dermatophytosis. However, we believe we would need to address multiple aspects and in multiple cities with a level of detail that goes beyond the scope of our metanalysis. However, we have decided to use this suggestion from the reviewer and mention it a future perspective to analyse, considering our results (line 253-257)

Reviewer 2 Report

Hedgehogs are known to be suitable hosts for a wide variety of parasites. The present manuscript presents a very cleverly done review on parasites with potential zoonotic risk. I have a few minor comments on the text.

1. line 64 - the correct title should be Borrelia burgdorferi sensu lato, with Borrelia burgdorferi italicized and sensu lato not italicized; the text incorrectly states ... latum ...

2nd line 70 - there should be a space between the round bracket and the square bracket; 

3.            Table 1 - should correctly read E. coli and not E.coli as indicated (i.e. with a space);

4.            Table 2 - I don't understand why the word Near is crossed out and underlined ?

5. line 120 - again, correctly it should be B. burgdorferi sensu lato and not as indicated ...lactum...;

6. line 128 - again, correctly should be E. coli and not as indicated E.coli (i.e. with a space);

7. section 3.3 - asks for in-text citations and also to state specifically what species of ticks and fleas parasitised the hedgehogs, the data given here is very general;

8. line 164 - should correctly read A. phagocytophilum and not A.phagocytophilum as given (i.e. with a space);

9. line 193 - correctly should be phagocytophilum and not phagocytophilium as in the text;

I recommend publication after these minor corrections.

Author Response

Dear Reviewer 1,

Thank you for your time reviewing our manuscript. Please find the answer to each one of your corrections/suggestions after each of the following paragraphs. 

Hedgehogs are known to be suitable hosts for a wide variety of parasites. The present manuscript presents a very cleverly done review on parasites with potential zoonotic risk. I have a few minor comments on the text.

We really appreciate your positive feedback regarding our manuscript.

  1. line 64 - the correct title should beBorrelia burgdorferisensu lato, with Borrelia burgdorferi italicized and sensu lato not italicized; the text incorrectly states ... latum ...We have corrected this accordingly. We also have corrected this expression in other parts of the manuscript when it appears.

2. line 70 - there should be a space between the round bracket and the square bracket; 

Corrected

  1. Table 1 - should correctly read E. coli and not E.coli as indicated (i.e. with a space);Corrected
  2. Table 2 - I don't understand why the word Near is crossed out and underlined ?Corrected
  3. line 120 - again, correctly it should beB. burgdorferisensu lato and not as indicated ...lactum...;Corrected
  4. line 128 - again, correctly should be E. coli and not as indicated E.coli (i.e. with a space);Corrected
  5. section 3.3 - asks for in-text citations and also to state specifically what species of ticks and fleas parasitised the hedgehogs, the data given here is very general;We have decided to add some in-text citations, considering where the information came from. We also provided some examples on the species of fleas and ticks to enrich this paragraph a little bit more. The final result is: 

    "More parasitic diseases have been reported in urban areas. Vectors (as fleas and ticks) and vector-borne parasites (Anaplasma phagocytophilum) had the highest prevalences, considering all the recorded pathogens. In Ulm (Germany), ticks (as Ixodes spp.) and fleas (as Archeopsylla erinacei) were present in 88% and 89% of the hedgehogs (n=56) [39]. A. phagocytophilum was assessed in 112 hedgehogs from three different cities in the Czech Republic with a very high prevalence (96%) [37]."

  6. line 164 - should correctly readA. phagocytophilumand not A.phagocytophilum as given (i.e. with a space);Corrected
  7. line 193 - correctly should bephagocytophilumand not phagocytophilium as in the text;Corrected

I recommend publication after these minor corrections.

Again, we would like to thank this reviewer for his/her time reviewing our manuscript and thank you for the recommendation.

Reviewer 3 Report

This study provide important epidemiologic information on hedgehogs zoonotic diseases and the manuscript is well structured. I suggest some minor revision.

Specific comments

Results

Table 1. Authors should report Giardia Assemblage and Cryptosporidum species reported by Krawczyk et al., 2015.

Discussion

Authors should also mention Giardia reported in rural location (33% of P).

Author Response

This study provide important epidemiologic information on hedgehogs zoonotic diseases and the manuscript is well structured. I suggest some minor revision.

We would like to thank Reviewer 2 for his/her time reviewing our manuscript. Please find answers to each one of your corrections/suggestions in the following paragraphs. Moreover, thank you for your positive feedback regarding our work.

Specific comments

Results

Table 1. Authors should report Giardia Assemblage and Cryptosporidum species reported by Krawczyk et al., 2015.

We have re-read the article by Krawczyk et al. 2015 and we have decided to leave the table without specifying the species of parasites, mainly because these authors do not specify the prevalences of each parasite species and, regarding Giardia, it is not clear that all the positive samples were confirmed as Giardia duodenalis. In fact, authors present a table of results with the same nomenclature as we do, so we do not want to take the risk and presenting incorrect information. However, we totally understand the reviewer's suggestion and we appreciate it.

Discussion

Authors should also mention Giardia reported in rural location (33% of P)

Following your suggestion, we have decided to mention the prevalence differences between rural and urban areas (line 154).

Again, thank you very much for reviewing our manuscript. 

Reviewer 4 Report

In this review, entitled "Do urban hedgehogs (Erinaceus europaeus) represent a relevant source of zoonotic diseases?", the authors analyze the role of the hedgehog (Erinaceus europaeus) as a reservoir of zoonotic diseases.

The study is well designed and conducted with scientific rigor, unfortunately the bibliography available at the moment is not so exhaustive on the topic and the available data are insufficient to fully understand the role of the hedgehog in a "One health" perspective. The merit of the authors is to clearly provide the information available to date and to propose, where possible, a comparison between the urban environment and the rural/wild environment in the countries where studies on these animals have been conducted. For these reasons, I would encourage the publication of this manuscript in "Pathogens".

However, It's well know that diseases increasingly emerge into animal and human populations as a consequence of the complex processes of interactions, often unseen, between wildlife,  human, vectors and the environment. In many countries of the world, urbanization causes dramatic changes in natural landscapes, favoring the adaptation of some wild animals to urban, peri-urban and rural habitats thanks to greater availability of the food supply. 

As for the hedgehog's diet, I would suggest editing and deepening line 54 of the manuscript. Hedgehogs make use of a wide variety of other available food items including eggs and nestlings of ground-nesting birds, mice and frogs, and food offered by humans. This last aspect is particularly important, as it causes greater interaction with human beings with the consequent health risks.

Author Response

In this review, entitled "Do urban hedgehogs (Erinaceus europaeus) represent a relevant source of zoonotic diseases?", the authors analyze the role of the hedgehog (Erinaceus europaeus) as a reservoir of zoonotic diseases.

At first, we would like to thank reviewer 3 for reviewing our manuscript. Please find answers to your comments/corrections/suggestions in the following paragraphs. 

The study is well designed and conducted with scientific rigor, unfortunately the bibliography available at the moment is not so exhaustive on the topic and the available data are insufficient to fully understand the role of the hedgehog in a "One health" perspective. The merit of the authors is to clearly provide the information available to date and to propose, where possible, a comparison between the urban environment and the rural/wild environment in the countries where studies on these animals have been conducted. For these reasons, I would encourage the publication of this manuscript in "Pathogens".However, It's well know that diseases increasingly emerge into animal and human populations as a consequence of the complex processes of interactions, often unseen, between wildlife,  human, vectors and the environment. In many countries of the world, urbanization causes dramatic changes in natural landscapes, favoring the adaptation of some wild animals to urban, peri-urban and rural habitats thanks to greater availability of the food supply. 

Thank you very much for the positive feedback regarding our work. We definitely agree that the literature is still scarce to allow a complete perspective on the role of hedgehogs as vectors and/or reservoirs of most zoonotic agents. However, we definitely believe our work summarises most published literature on the subject, working as a state-of-the-art for future projects and assessments. We also definitely agree that urban wildlife is gaining attention and more research needs to be done considering these populations to understand infectious disease dynamics in urban environments.

As for the hedgehog's diet, I would suggest editing and deepening line 54 of the manuscript. Hedgehogs make use of a wide variety of other available food items including eggs and nestlings of ground-nesting birds, mice and frogs, and food offered by humans. This last aspect is particularly important, as it causes greater interaction with human beings with the consequent health risks.

We agree. Considering reviewer's suggestion, we have decided to modify the Introduction paragraph where we describe hedgehogs' diet, emphasizing the close contact with people (new lines 62-64).

Again, we would like to thank reviewer 3 for the positive and constructive feedback regarding our manuscript. 

Round 2

Reviewer 1 Report

I recommend publication after the changes and future perspectives indicated by the authors.